# Intraspecific venom variation in the medically important puff adder (*Bitis arietans*): Comparative venom gland transcriptomics, *in vitro* venom activity and immunological recognition by antivenom

**Charlotte A. Dawson**[1,2]*, **Keirah E. Bartlett**[1,2], **Mark C. Wilkinson**[1,2], **Stuart Ainsworth**[1,2,3], **Laura-Oana Albulescu**[1,2], **Taline Kazandijan**[1,2], **Steven R. Hall**[4], **Adam Westhorpe**[1,2], **Rachel Clare**[1,2], **Simon Wagstaff**[2], **Cassandra M. Modahl**[1,2,5], **Robert A. Harrison**[1,2], **Nicholas R. Casewell**[1,2]

1 Centre for Snakebite Research and Interventions, Liverpool School of Tropical Medicine, Liverpool, United Kingdom, 2 Department of Tropical Disease Biology, Liverpool School of Tropical Medicine, Liverpool, United Kingdom, 3 Department of Infection Biology and Microbiomes, Institute of Infection, Veterinary and Ecological Sciences, University of Liverpool, Liverpool, United Kingdom, 4 Department of Biomedical and Life Sciences, Lancaster University, Lancaster, United Kingdom, 5 Department of Vector Biology, Liverpool School of Tropical Medicine, Liverpool, United Kingdom

* Charlotte.Dawson@lstmed.ac.uk

## Abstract

### Background

Variation in snake venoms is well documented, both between and within species, with intra-specific venom variation often correlated with geographically distinct populations. The puff adder, *Bitis arietans*, is widely distributed across sub-Saharan Africa and into the Arabian Peninsula where it is considered a leading cause of the ~310,000 annual snakebites across the region, with its venom capable of causing substantial morbidity and mortality. Despite its medical importance and wide geographic distribution, there is little known about venom variation between different *B. arietans* populations and the potential implications of this variation on antivenom efficacy.

### Methodology

We applied a range of analyses, including venom gland transcriptomics, *in vitro* enzymatic assays and reverse phase chromatography to comparatively analyse *B. arietans* venoms originating from Nigeria, Tanzania, and South Africa. Immunological assays and *in vitro* enzymatic neutralisation assays were then applied to investigate the impact of venom variation on the potential efficacy of three antivenom products; SAIMR Polyvalent, EchiTAb-Plus and Fav-Afrique.

**Data Availability Statement:** All relevant data are within the manuscript and supporting files.

Sequences produced through transcriptomic analyses are available through GenBank, NCBI (Accession numbers PP950449 - PP950655).

**Funding:** UK Medical Research Council Doctoral Training Programme PhD studentship to CAD. Work was also supported by UK Medical Research Council Grant (MR/L01839X/1) awarded to RAH and NRC. The funders had no role in study design, data collection and analysis, decision to publish, or preparation of the manuscript.

**Competing interests:** The authors have declared that no competing interests exist.

### Findings

Through the first comparison of venom gland transcriptomes of *B. arietans* from three geographically distinct regions (Nigeria, Tanzania, and South Africa), we identified substantial variation in toxin expression. Findings of venom variation were further supported by chromatographic venom profiling, and the application of enzymatic assays to quantify the activity of three pathologically relevant toxin families. However, the use of western blotting, ELISA, and *in vitro* enzymatic inhibition assays revealed that variation within *B. arietans* venom does not appear to substantially impact upon the efficacy of three African polyvalent antivenoms.

### Conclusions

The large distribution and medical importance of *B. arietans* makes this species ideal for understanding venom variation and the impact this has on therapeutic efficacy. The findings in this study highlight the likelihood for considerable venom toxin variation across the range of *B. arietans*, but that this may not dramatically impact upon the utility of treatment available in the region.

### Author summary

The puff adder (*Bitis arietans*) is found across sub-Saharan Africa and the Arabian Peninsula and is capable of causing life threatening pathology due to its potent venom. The extensive range of *B. arietans* exposes populations to different ecological pressures which may impact upon the composition of venom toxins. In this study, we examined the venom composition of *B. arietans* from three countries separated by large geographic distance: Nigeria, Tanzania and South Africa. By integrating venom gland transcriptomes, venom chromatography, and *in vitro* functional assays to profile *B. arietans* venom composition, we uncovered extensive variation between the three locales. Given that venom variation can have a significant impact on the efficacy of antivenom treatment, we also investigated the ability of three African antivenoms to recognise and inhibit *in vitro* venom activity. Through these analyses, we were able to determine that venom variation did not have a substantial impact on the neutralising effect of selected antivenoms. This study has highlighted the potentially extensive venom variation found across the range of *B. arietans* and initiated valuable investigations into the efficacy of African antivenoms to protect human populations vulnerable to snakebite envenoming.

### 1. Introduction

The composition and function of venoms have been refined through millions of years of evolution in response to selective pressures associated with specific ecological niches [1–6]. Snake venoms are no exception, and their composition is thought to be honed by various factors including prey availability, life stage, available ecological niches, and the need for self-defence [2–6]. Variation between venoms is well documented at different taxonomical levels; between taxonomic families, species, and even intraspecifically [4,5,7–10].

Throughout its wide range across much of sub-Saharan Africa and parts of the Arabian Peninsula, the puff adder (*Bitis arietans*) is considered one of the most medically important snake species [11–13]. Puff adder venom is potently cytotoxic, causing rapid onset of pain, swelling, extensive blistering, inflammation, and necrosis [12–14]. These local manifestations can leave patients vulnerable to secondary infections, which themselves can prove fatal without prompt and appropriate treatment [15]. Systemically, the venom can also elicit potent haemotoxicity, manifesting as hypotension, spontaneous haemorrhage, thrombocytopenia, and coagulopathy [12,13,16,17]. Bites from *B. arietans* contribute substantially towards the estimated 314,078 annual envenomings across sub-Saharan Africa [11–13].

The mortality and morbidity observed following snakebite envenoming is the result of pathology caused by the diverse repertoire of toxins that make up venom. The predominant toxin families within *B. arietans* venom are snake venom metalloproteinases (SVMPs), snake venom serine proteases (SVSPs), phospholipase $A_2$s (PLA$_2$s) and C-type lectin-like proteins (CLPs), with SVMP and SVSPs accounting for up to 50% of dried venom weight [5,18–22]. The activity of these toxins underlies the observed pathology in snakebite victims, including haemorrhage, coagulopathy, and necrosis [23–25]. For example, the SVSP KN-Ba, has been demonstrated to cleave the α- and β-chains of fibrinogen and induce inflammation through the production of inflammatory mediators such as tumour necrosis factor (TNF), Interleukin-6 (IL-6) and Interleukin-1β (IL-1β) [18,26]. Numerous CLPs have also been isolated from *B. arietans*, one example being Bitiscentin-3, which has been demonstrated to cause platelet agglutination and bind von Willebrand Factor [27,28]. *Bitis* venoms are also rich in short peptides [29]. Two classes of these, proline-rich oligopeptides (PRO) and bradykinin-potentiating proteins (BPPs), are potent hypotensive peptides and may contribute towards the cardiovascular effects observed following envenoming [29].

Although prior studies have provided valuable insight into puff adder venom toxin composition [5,7,22,30,31], the wide geographical range of this species provides great potential for intraspecific venom variation. Currier *et al.* demonstrated extensive variation between different *B. arietans* geographic localities when venoms were profiled via SDS-PAGE gel electrophoresis and zymographic examination of protease activity [5]. Nigerian *B. arietans* displayed significantly greater protease-mediated degradation of gelatin and fibrinogen than samples from Ghana, Malawi, Tanzania, Zimbabwe, and Saudi Arabia [5]. More recently, research focusing on *B. arietans* venom collected from two different Angolan provinces also showed intra-specific variation in venom function, with samples from the Calandula region having significantly greater PLA$_2$ activity than those in the Mufuam region [22].

The consequences of venom variation at a species level can have important implications for the clinical management of human snakebite victims [32]. Intraspecific venom variation can also have a significant impact on the efficacy of antivenom. For example, preclinical studies have shown that commercially available antivenoms have variable efficacies at neutralising different regional populations of the Indian cobra (*Naja naja*) [33] and the common lancehead (*Bothrops atrox*) in Brazil [34]. However, a study examining the efficacy of EchiTAb-Plus antivenom, which is made against multiple snake species including *B. arietans*, showed there was no significant difference in the antivenom dose required to prevent lethality caused by Nigerian and Cameroonian *B. arietans* venom in a murine model of envenoming [35].

Nevertheless, the finer details of venom variation found across the geographical range of *B. arietans* remains largely uncaptured, and the potential clinical implication of such variation is not fully understood. To address these limitations, in this study we performed comparative transcriptomic analyses of the toxin genes expressed in the venom glands of *B. arietans* snakes sourced from three different geographic locales, Nigeria, Tanzania, and South Africa. We supplemented these data with compositional and functional comparative analyses using venom

sourced from a pool of *B. arietans* specimens from same regions, including those used for tran-scriptomic analyses, and then explored the ability of available antivenoms directed against *B. arietans* to bind to and neutralise these venom activities. Our results show that *B. arietans* from three geographic regions have distinct venom gland transcriptomes, venom proteomes, and functional differences in venom composition, but this does not have a major impact on the *in vitro* binding and neutralising effect of three polyvalent antivenoms.

## 2. Methods

### 2.1 Venoms and antivenoms

Venoms were extracted from specimens of *B. arietans* maintained in the herpetarium of the Centre for Snakebite Research and Interventions at the Liverpool School of Tropical Medicine, UK. Venoms from each locale were pooled and lyophilised (six animals from Nigeria, three from Tanzania and one from South Africa) for long term storage at 2–8 ˚C. Venoms were reconstituted to 10 mg/mL in sterile PBS (pH 7.4, Gibco, 10010–015) ahead of experimental use and stored at -20 ˚C short term. Three medically relevant polyvalent equine antivenoms produced for use in sub-Saharan Africa were used in this study: EchiTAb-Plus (ICP, lot: 5370114PALQ, expiry: 01/2017), SAIMR Polyvalent (SAVP, lot: BF00546, expiry: 11/2017), and Fav-Afrique (Sanofi Pasteur, lot: K8453, expiry: 06/2016). EchiTAb-Plus was supplied as intact Ig, raised against Nigerian *Echis ocellatus*, *Naja nigricollis* and *Bitis arietans*. SAIMR Polyvalent antisera was raised against 10 medically important species from sub-Saharan Africa (*Hemachatus haemachatus*, *Dendroaspis polylepis*, *D. jamesonii*, *D. angusticeps*, *N. nivea*, *N. melanoleuca*, *N. mossambica*, *N. annulifera*, *B. gabonica* and *B. arietans*). Fav-Afrique antisera was raised against 10 medically important species (*E. ocellatus*, *E. leucogaster*, *N. nigricollis*, *N. melanoleuca*, *N. nigricollis*, *D. polylepis*, *D. viridis*, *D. jamesoni*, *B. gabonica* and *B. arietans*). Both SAIMR Polyvalent and Fav Afrique were supplied as $F(ab')_2$ products. All three products were used in liquid formats (10 mL per vial). Expired antivenoms used in this study were donated by Public Health England.

### 2.2 RNA isolation, sequencing, and bioinformatics

Three specimens of *B. arietans* (one from Nigeria, one from Tanzania and one from South Africa) were humanely euthanised using an overdose of injectable Pentojet (pentobarbital sodium) three days after venom extraction. Venom glands were immediately dissected, and flash-frozen in liquid nitrogen. Thereafter, venom gland tissue (50–200 mg) was ground in liq-uid nitrogen using a pestle and mortar and resuspended in 1mL TRIzol (Ambion, 155169026) before the addition of 200 µL of chloroform (Sigma, 288306). Samples were then centrifuged at 12,000 x *g* for 15 minutes at 4 ˚C, and the resulting aqueous phase transferred to a sterile 1.5 mL centrifuge tube and mixed 1:1 with 70% ethanol. Samples were then transferred to a RNeasy spin column (Qiagen, 74134) in 700 µL aliquots and spun at 12,000 x *g* for 15 minutes. RNA purification was performed using a Dynabeads mRNA DIRECT purification kit protocol (ThermoFisher, CAT: 61011) as per supplier instructions. 100 µL of purified RNA was added to 1 mg of prepared Dynabeads (75 µg RNA per 200 µL beads). Beads were incubated for 3–5 minutes at RT before being magnetically separated. The beads were then resuspended in 10 µL of elution buffer (10 mM Tris-HCl, pH 7.5) and heated to 65 ˚C for 2 minutes, before eluting the captured RNA. A second round of PolyA selection was then performed, using the same beads used in the first round. Bound RNA was eluted as previously, via the addition 10 µl of elution buffer and heating to 65 ˚C for 2 minutes.

RNA-seq library preparation and sequencing were performed commercially at the Centre for Genomics Research, University of Liverpool. For library preparation, 50 ng of purified

mRNA and the Script-Seq v2 RNA-Seq Library preparation kit was used according to manufacturer's protocol (epicentre), with 12 cycles of amplification, before purification using AMPure XP beads (Fisher Scientific, 10136224) and quantification via Qubit dsDNA HS Assay kit (Thermo, Q32851). Size distribution of the libraries was determined using a Bioanalyser (Agilent). Generated sequencing libraries were multiplexed and sequenced using an Illumina MiSeq instrument (1/6[th] of a lane per transcriptome), with 2 x 250 bp pair-end sequencing technology. The three samples from this study were sequenced on a single lane in multiplexed fashion.

Resultant raw data was quality controlled through the removal of any adapter sequences using Cutadapt [36] and trimming low quality bases using Sickle [37]. Following this initial quality control step, reads were next *de novo* assembled into contigs for each animal using the assembly program VTBuilder [38] under the following parameters: minimum transcript length of 150 bp and a minimum isoform similarity 96%. Additionally, reads were *de novo* assembled with Extender [39] using 10,000 starting seeds, where seeds were reads that had first been merged with PEAR (Paired-End read mergeR; v0.9.6 using default parameters) [40] and seed extensions required 100 nucleotide overlaps and quality scores of at least 30.

*De novo* assemblies for each *B. arietans* venom gland transcriptome were merged, and redundancy and contigs less than 150 bp were removed with CD-HIT (v4.8.1) [41,42]. Toxins were then annotated using ToxCodAn [43] and Diamond (v2.0.11) [44] BLASTx (E-value $10^{-05}$ cut-off) searches against the National Center for Biotechnology Information (NCBI) non-redundant protein database (accessed February 2023). Toxin transcripts were manually checked to determine that all translations were non-redundant, full-length proteins (methionine start codon to stop codon), had a maximum of three ambiguous amino acid residues, shared sequence identity with currently known toxins, and contained a conserved signal peptide sequence within each venom protein family. Abundances of transcripts from the final annotated toxin set were determined with RSEM (RNA-seq by Expectation Maximization, v1.3.10) [45], using the aligner Bowtie2 (v2.3.5) [46]. Chimeric sequences were identified using ChimeraKiller v 0.7.3 [47]. Potential chimeric contigs were then cross checked against proteomics data [48] and sequences curated as required.

## 2.3 RP-HPLC analysis

Reverse phase HPLC (RP-HPLC) was performed on a Vanquish HPLC system (Thermo) using a Biobasic C4 column (2.1 x 150 mm, Catalog number: 72305–051030). The three venom samples were prepared by adding trifluoroacetic acid (TFA) at a final concentration of 0.1% (v/v) to a pre-clarified 0.5 mg/mL venom solution and then centrifuging at 10,000 x *g* for 5 mins. A 100 µL aliquot of this solution was injected onto the column. The flow rate was 0.2 mL/min and proteins were separated using the following gradient of acetonitrile in 0.1% TFA: 0–25% over 5 mins, 25–55% over 50 mins, then 55–80% over 10 mins. The separation was monitored at 214 nm. Fractions were also subjected to SDS-PAGE under non-reducing and reducing conditions. The gels used were BioRad 4–20% acrylamide, stained with Coomassie Blue R250.

## 2.4 Enzymatic assays

To ascertain the impact of intraspecific variation on the functional activity of the selected venoms, a panel of *in vitro* assays were applied. The SVMP activity of the three venom samples was determined using a kinetic fluorescent assay with the SVMP-specific fluorogenic substrate ES010 (R&D Biosystems) [49]. Venoms were diluted to 1 mg/mL in PBS, and 1 µL added per well to a 384-well plate (Greiner, 781101), with all samples tested in triplicate. ES010 substrate

(6.2 mM in DMSO) was diluted in reaction buffer (150mM NaCl, 50 mM Tris-HCl pH 7.5) before 75 μL of substrate was added to each well, giving a final reaction concentration of 7.5 μM. Plates were read using an Omega ClarioStar (BMG) at 25 ˚C for 1 hour, using an excitation wavelength of 320 nm and an emission wavelength of 405 nm and set for 25 cycles (154 seconds per cycle) at 20 flashes per well. For statistical analysis, fluorescence at the 1-hour timepoint was used to perform a Two-Way ANOVA with Tukey's multiple comparison in GraphPad Prism 9.

Venom PLA$_2$ activity was measured using a commercially available Abcam colorimetric sPLA2 assay kit (ab133089). Venoms were diluted to 10 ng/μL and then 1 μl was transferred per well to a 384-well plate, followed by 5 μL of kit-supplied DTNB mixture (10mM, in 0.4 M Tris-HCl, pH 8) and 30 μL of the supplied substrate (Diheptanoyl Thio-PC, 1.66 mM). Plates were immediately transferred to a ClarioStar Omega plate reader (25 ˚C, 405 nm, 6 cycles, 11 flashes per cycle, 161 seconds per cycle). Specific PLA$_2$ activity (μmol/min/mL) was calculated using the following equation:

$$sPLA2\ Activity(umol/min/mL) = \frac{\Delta A_{414}/\min X\ 0.255\ mL}{10.66\ mM^{-1}} x\ \frac{Sample\ Dilution}{0.01\ mL} \tag{1}$$

Statistical significance between resulting PLA$_2$ activities was analysed via Two-Way ANOVA, with Tukey's multiple comparison test, using GraphPad Prism 9.

Serine protease (SVSP) activity was determined using an absorbance-based Chromogenic S-2288 assay (Chromogenix, 82085239). Venoms were diluted to 0.066 mg/mL and 15 μL added per well in quadruplicate to a 384-well plate before incubation for 3 mins at 37 ˚C. 15 μL of assay buffer was added per well, before incubation at 37 ˚C for a further 3 mins. Assay substrate was diluted to 6 mM in ddH$_2$O and, following the final incubation step, 15 μL/well of substrate was added and the plate immediately read using an Omega ClarioStar plate reader (405 nm, 37˚C, 80 cycles). Using kit instructions, activity was calculated and displayed as ukat/I and U/I values. Statistical significance was calculated using a Two-way ANOVA with Tukey's multiple comparison test, using GraphPad Prism 9.

## 2.5 MTT cell viability assays

Cytotoxic activity of the venoms was measured using MTT (3-[4,5-dimethylthiazol-2-yl]-2,5-diphenyltetrazolium bromide) cell viability assays [50]. HaCaT immortalised human keratinocytes (Caltag Medsystems; Buckingham, UK) were seeded at 20,000 cells/well in 96-well plates in Dulbecco's Modified Eagle Medium (DMEM; Gibco, 61965026), supplemented with 10% (v/v) foetal bovine serum (FBS; Thermo Fisher Scientific, 26140079), 2% (v/v) sodium pyruvate (Fisher Scientific Ltd, 113600399), and 1% (v/v) penicillin/streptomycin (Fisher Scientific Ltd, 15140122), and incubated overnight at 37˚C, 5% CO$_2$ and 85% humidity–hereafter referred to as standard conditions. Wells were treated in triplicate with 100 μL of *B. arietans* crude venom from either Nigeria (13.6–72.6 μg/mL), Tanzania (8.6–37.0 μg/mL), or South Africa (5.0–110 μg/mL). After 24 h incubation under standard conditions, Thiazole Blue tetrazolium bromide (MTT; M5655, Sigma Aldrich) solution (120 μL at 0.83 mg/mL) was added to all wells and the plates were incubated for 45 min in standard conditions. Thereafter, cell medium was aspirated, and 100 μL DMSO was added to the wells to dissolve the biosynthesised formazan crystals and absorbance (550 nm) was read for all wells using a BMG Clariostar plate reader. Experiments were completed independently in triplicate for each venom, where each n was conducted at a time independent of all other matched n's. All venoms within an 'n' were completed in triplicate wells and the mean taken as the final value for that one trial. Subsequently, data were plotted as dose-response curves using GraphPad Prism 9.

## 2.6 Quantitative western blotting

Venom toxin recognition by antivenom was determined using quantitative western blotting. 4 µg of venom in 10 µL of reducing SDS-PAGE buffer (6X stock solution, NEB B7024S with 30% β-mercaptoethanol) was run per lane on BioRad Mini PROTEAN pre-cast TGX acrylamide gels (15 well, any kDa, 4569036) alongside 5 µL of protein marker (Thermo Scientific, PageRuler Prestained, 26616). Gels were run at 160 V for 1 hour before being transferred to 0.2 µM nitrocellulose membranes (Bio-Rad Trans-Blot Turbo Mini, 1704158) via a BioRad Trans-Blot Turbo Transfer system (7-minutes, 2.5 V). Proteins were visualised using Revert total protein stain (Li-Cor, 92611011) at 700 nm to ensure full transfer to nitrocellulose membranes. Membranes were blocked for two hours in blocking buffer (5% powdered milk in TBST (10 mM Tris-Cl pH 8.5, 150 mM NaCl 0.1% TWEEN-20). Thereafter, antivenoms were standardised to 10 mg/mL and diluted 1:1000 in blocking buffer, before being incubated with membranes overnight at 2–8 ˚C. Membranes were washed in TBST prior to the addition of secondary antibodies (Fluorescently conjugated DyLight800 rabbit anti-horse, Cambridge Bioscience, 608445002, 1:15,000). Secondary antibody was incubated with membranes for 2 hours at RT in PBS. Secondary antibodies were then removed, and membranes were washed with TBST. Membranes were imaged using a LiCor Odyssey system (800 nm for 2 minutes, 700 nm for 2 minutes). Fluorescence signal was normalised according to the provided LiCor instructions. Normalised signals were plotted as mean ± SD in using GraphPad Prism 9.

## 2.7 ELISA evaluation of antivenom binding

ELISA plates (96-well, ThermoFisher, 3655) were coated with 100 ng/well of each venom diluted in carbonate-bicarbonate buffer (50 mM, pH 9.6, Sigma, C3041) and incubated for 1 hour at 37 ˚C. Coating buffer was discarded, and plates were washed through the addition and discarding of 200 µL TBST per well, repeated six times. Post washing, 100 µL of blocking buffer was added per well and plates incubated for 1 hour at 37 ˚C, before a second round of wash steps. Neat antivenom (EchiTAb-Plus, SAIMR Polyvalent and Fav-Afrique) were diluted 1:100 in blocking buffer and 125 µL of diluted antivenom was then transferred to the ELISA plates before being serially diluted 1:4 across the plate in blocking buffer. Plates were incubated at 2–8 ˚C overnight and then washed again in TBST. Thereafter, rabbit anti-horse IgG conjugated to horseradish peroxidase (Sigma, A6917-1mL) was diluted 1:2000 in PBS and 100 µL added per well. The plates were incubated for 2 hours at room temperature in the dark before a final wash step with TBST ahead of substrate addition. A 100 mg tablet of 2,2′-Azino-bis (3-ethylbenzothiazoline-6-sulfonic acid) diammonium salt (ABTS, Sigma A9941) was dissolved in 100 mL of citrate buffer (50 mM, pH 4) containing 25 µL of $H_2O_2$ (Sigma, H1009). 100 µL of the resulting buffer was added per well and allowed to develop for 15 minutes in the absence of light, prior to the addition of 50 µL of 1% SDS (Severn Biotech). Results were recorded by measuring well absorbance (405 nm) using a FluoSTAR Omega spectrophotometer (BMG LabTech), with data exported to Microsoft Excel for normalisation by subtracting background absorbance readings. Data was then exported to GraphPad Prism 9, where the mean and SD of each data set was plotted. Differences in the binding levels of the three antivenoms tested were then determined using a Two-way ANOVA, with Tukey's multiple comparison test.

## 2.8 *In vitro* inhibition of venom activity

Antivenom was applied to all three enzymatic toxin assays to assess the inhibition of toxin activity by undiluted antivenom. SVMP inhibition was determined via incubation of 1 µg of venom with 15 µL of each antivenom of interest. Venoms and antivenoms were incubated for

30 minutes at 37 ˚C before adding substrate and reading the plate as detailed above. PLA$_2$ inhibition was measured as follows: 9 μL of undiluted antivenom was incubated with 1 μL of 20 ng/μL venom for 30 minutes at 37 ˚C. Following incubation, 5 μL of DTNB was added per well followed by 30 μL of substrate and assay buffer. Plates were then immediately transferred to a ClarioStar Omega plate reader and read as described above. Finally, SVSP inhibition was determined via incubation of 15 μL of undiluted antivenom with 1 μg of venom for 30 minutes at 37 ˚C. Following incubation, substrate was added, and plates measured via a ClarioStar Omega plate reader as detailed previously. For all assays, percentage inhibition of toxin activity was then calculated relative to the venom-only controls. Percentage inhibition was then plotted (mean ± SD) in GraphPad Prism 9. Statistical differences in inhibition between different geographic localities was calculated via two-way ANOVA with Tukey's multiple comparison test.

## 3. Results

### 3.1 *Bitis arietans* venom gland transcriptomes exhibit intraspecific variation in venom toxin expression

The total number of quality-controlled sequence reads for the Nigerian, Tanzanian and South African venom gland transcriptomes were 3,951,423, 2,455,567, and 2,196,166, respectively. Within those datasets, 77, 68, and 68 contigs corresponded to full length venom toxins from Nigeria, Tanzania, and South Africa, respectively. Chimeric contig analysis identified 12 sequences with poor coverage (1, 5 and 6 from Nigeria, Tanzania and South Africa respectively) which were then compared against proteomic analyses conducted by Wilkinson *et al* [48]. Proteomic expression was found for one sequence (accession number PP950538) and remaining sequences were removed resulting in a final data set of 71, 64 and 62 contigs from Nigeria, Tanzania and South Africa respectively. Sequenced reads were deposited onto NCBI (BioProjectID: PRJNA1060480, accession numbers PP950449—PP950655). From this point onward, sequences and their relative expression will be discussed with regards to the "toxinome"–the total expression level of venom toxin contigs.

There were 16 different toxin families present in all three venom gland localities sequenced, with six to seven toxin families per locale comprising 90% of total toxin expression (Fig 1a) and nine families comprising between 97–99% of the total toxinome (Fig 1b). A further nine toxin families made up remaining 1–3% of toxinome expression, with one of these families (CRISP) only found in the South African and Tanzanian datasets. Across all three toxinomes, the most abundantly expressed toxin family was the C-type lectin-like proteins (CLPs), which accounted for 32.9%, 33.5% and 29.5% of total contig expression (Nigeria, Tanzania, and South Africa respectively). A total of 33 distinct CLP transcripts were identified, with 17 contigs within the South African data set, correlating with the lower total abundance of CLPs in this toxinome. Whilst the Tanzanian and Nigerian data sets had comparable CLP expression (33.5% and 32.9% respectively), the former had a greater contig repertoire. A total of 33 contigs were identified within the Tanzanian data sets compared to the 22 detected in the Nigerian toxinome.

Within viperid venoms, one of the most pathologically important and abundant toxin families is that of the SVMPs [51]. Differences were observed in the expression levels of this toxin family between the three toxinomes. SVMPs were the second most abundant toxin group in the Tanzanian venom gland transcriptome (20.9%), whereas they were the third most abundant group in the Nigerian and South African datasets (14.9% and 12.7% respectively). The number of contigs in each data set also reflected the abundance data, with eight full length contigs assembled from the Tanzania data, six from the South African, and five from the Nigerian.

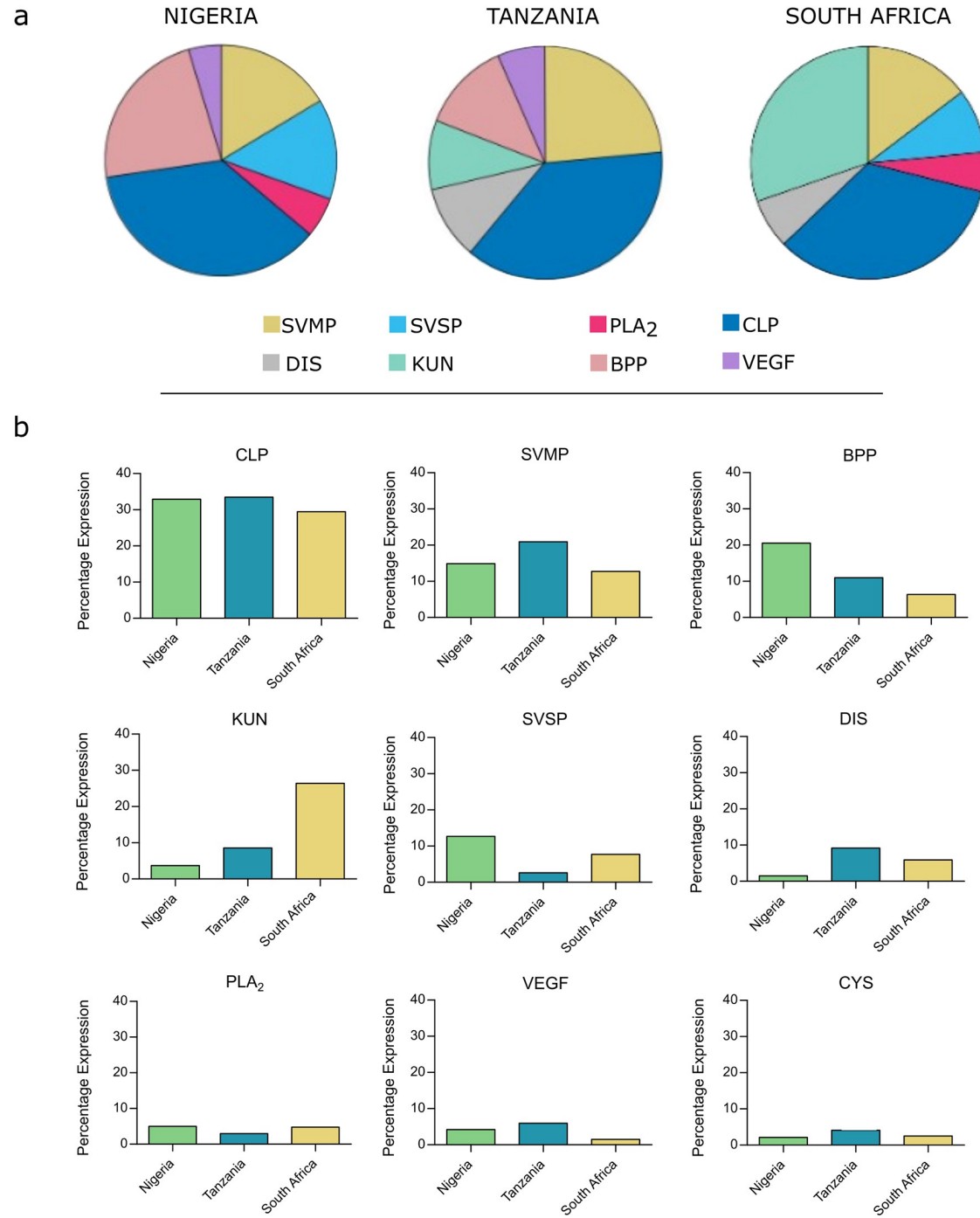

**Fig 1. Percentage expression of toxin families found in the venom gland transcriptomes of *Bitis arietans* from three different geographic locales.** a) Percentage abundance, calculated from relative toxin abundance data, of toxin families that constituting 90% of Nigerian, Tanzanian and South African *B. arietans* venom gland transcriptomes shown as pie-charts (produced using Canva). b) Side-by-side relative abundances (percentage expression of the toxinome) of detected toxins comprising ≥97% of the venom gland transcriptome and Nigerian. Tanzanian and South African *B. arietans*, displayed as bar charts. Snake venom metalloproteinases (SVMP), snake venom serine proteinases (SVSP), Phospholipase A$_2$s (PLA$_2$), C-type Lectin-like proteins (CLP), disintegrins (DIS), Kunitz-like proteins (KUN), Bradykinin potentiating peptide (BPP), Vascular endothelial growth factor (VEGF), and cystatin (CYS).

SVMP content can be further broken down based on structural sub-classes (i.e. PI, PII and PIII sub-classes) [52]. Examining the assembled contigs showed the presence of PII and PIII SVMPs, but an absence of PI SVMPs in all three *B. arietans* samples. Whilst P-II and P-III sequences were present in all datasets, the relative expression varied depending on locale. Both Tanzanian and Nigerian *B. arietans* had a greater proportion of their toxinomes attributable to PII SVMP expression compared to the South African *B. arietans* (17.8%, 12.4% and 5.0% respectively). PIII SVMPs expression within the South African toxinome was comparable to PII expression levels, at 3.6%. For both Nigerian and Tanzanian *B. arietans*, however, there was substantially lower expression of PIII SVMPs, at 2.5% and 1.1% respectively.

Intraspecific variation was also observed in the PLA$_2$ toxin family. All three venom gland transcriptomes contained three PLA$_2$ contigs. However, Nigerian *B. arietans* had a greater percentage of the total toxinome attributed to PLA$_2$ expression compared to both Tanzanian and South African datasets (5.0% compared to 3.0% and 4.8%, respectively). All contigs identified were predicted to encode for catalytically active PLA$_2$s, as evidenced by the presence of the D49 residue [53].

The expression levels of SVSP encoding contigs was, as with the previous toxin families, variable between the three localities studied. Nigerian *B. arietans* had the greatest percentage of SVSP expression at 12.7%, followed by South Africa at 7.7%, and Tanzania having the lowest percentage at 2.6%. The number of full-length contigs assembled followed the same trend, with Nigeria having the greatest number, followed by South Africa, and Tanzania with the least (24, 18 and 6 respectively).

It was of interest to note the percentage of the toxinome attributed to BPP. In the Nigerian venom gland transcriptome, these hypotensive peptides were the second-highest expressed toxin family, accounting for 20.5% of total toxin expression. Expression within both the Tanzanian and South African venom glands was considerably lower than that seen in the Nigerian toxinome, at 11.0% and 6.4% respectively.

## 3.2 Reverse phase chromatography supports transcriptomic evidence of intraspecific venom variation

Reverse phase chromatography (RP-HPLC) was used to determine whether the transcriptomic variation in toxin expression was also reflected in secreted venom proteins (Fig 2). Given their clear profiles, only the percentage abundance of two toxin families, CLPs and SVMPs, could be confidently quantified. The relative abundance of CLPs closely matched the transcriptomic expression levels for all three *B. arietans* locales (30.2–32.8% protein abundance, 29.5–33.5% transcriptomic expression). Comparing the percentage of SVMP expression at transcript and protein level showed a greater degree of discrepancy. Protein abundance was greater than the determined transcript abundance for all venoms; however, the variation seen between the three locales remained consistent, with Tanzanian *B. arietans* having the greatest SVMP expression at both transcriptomic and proteomic levels (20.9% transcript expression, 32.7% protein abundance), followed by South African and Nigerian *B. arietans* (South Africa; 12.7% transcriptome expression, 22.5% toxin abundance, Nigeria; 14.9% transcriptome expression, 18.6% toxin abundance).

## 3.3 *Bitis arietans* venoms show considerable intraspecific variability in functional activity

To determine whether intraspecific variation observed at both the transcriptomic and proteomic levels were reflected in enzymatic toxin activities, *in vitro* assays specific to three major families of venom toxins were conducted (Fig 3). SVMP activity was measured via a kinetic

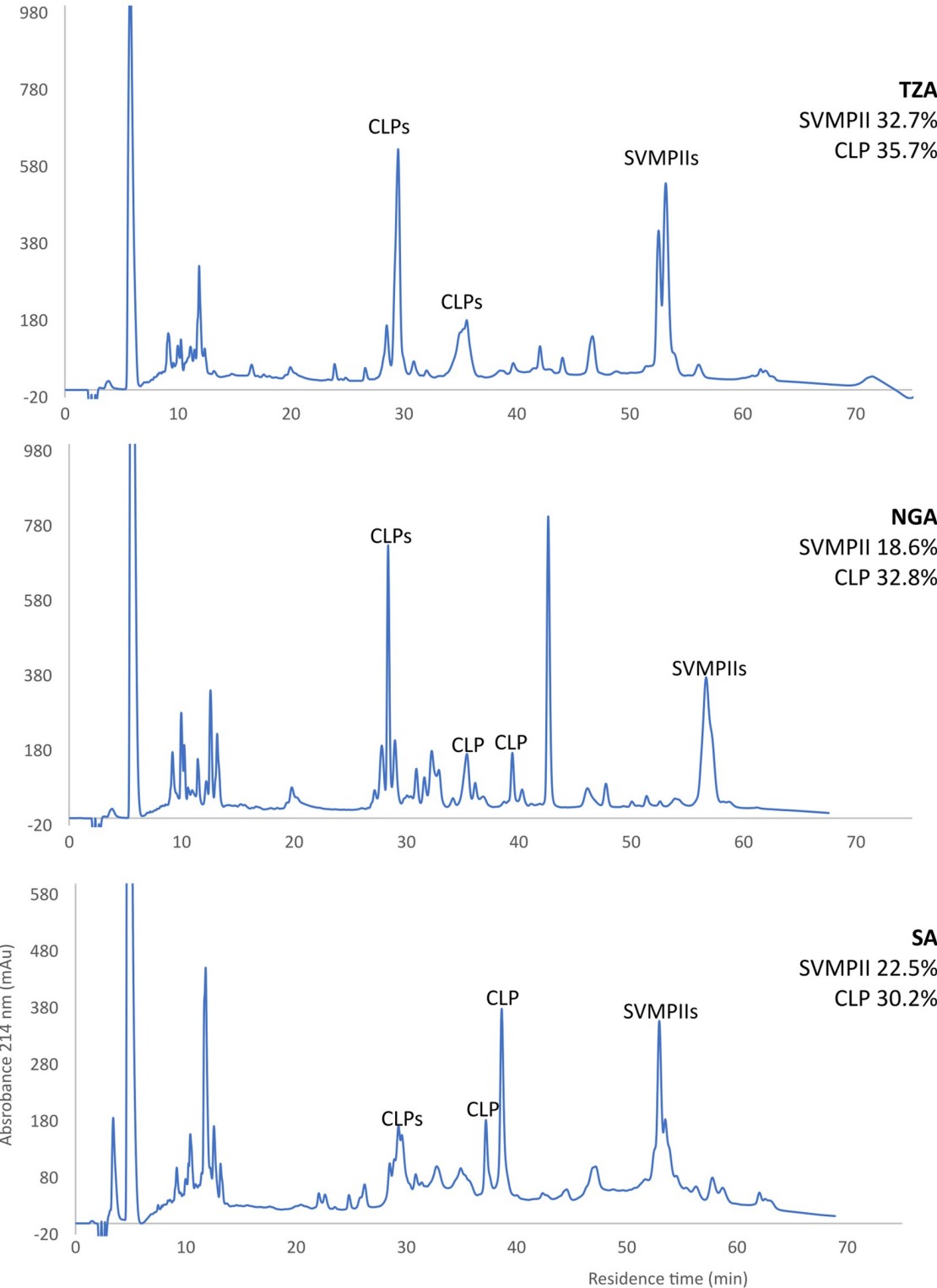

**Fig 2. RP-HPLC profiles of *Bitis arietans* venoms from three different geographical locales.** Whole venom was dissolved in PBS and a 30 μg sample was applied to a BioBasic C4 RP-HPLC column. The proteins were separated in a gradient of acetonitrile in SDS-PAGE sample buffer. The CLPs were identified by their characteristic electrophoretic patterns: paired bands of alpha and beta subunits at 14–18 kDa under reducing conditions, but a single band at 30–32 kDa under non-reducing conditions (S1 Fig). The position of the SVMPs was determined by comparison with that of *B. arietans* SVMPs purified and identified in a separate study [52].

Relative toxin percentages for SVMPs and CLPs are shown as a percentage of whole venom and were calculated from integrated peak areas.

fluorometric-substrate cleavage based enzymatic assay and revealed significant differences in venom SVMP activity between the three *B. arietans* localities. Tanzanian venom had the greatest SVMP activity (RFU 427,415 ± 16,396), followed by Nigerian *B. arietans* venom (RFU 390,155 ± 25,489) and the lowest activity was seen in the South African venom (150,895 ± 36,699). The extent of SVMP activity measured in both Nigerian and Tanzanian venom samples was significantly greater than in the South African *B. arietans* venom ($P < 0.0001$ for both comparisons) (Fig 3a).

Quantification of catalytic $PLA_2$ activity correlated with the assembled Nigerian *B. arietans* toxinome, with this locale having the greatest $PLA_2$ activity (90.2 ± 7.5/µg venom) (Fig 3b). Despite the similarity in transcriptomic $PLA_2$ abundance in Nigerian and South African *B. arietans*, the latter had significantly lower $PLA_2$ activity, 50.5 ± 4.8/µg venom ($P < 0.0001$). Tanzanian *B. arietans* had significantly greater activity (86.0 ± 0.9/µg venom) than South African *B. arietans* ($P < 0.0001$), despite having the lowest transcriptomic abundance.

The final enzymatic assay conducted measured SVSP activity via an absorbance-based chromogenic assay (Fig 3c). The quantification of the resulting enzymatic activity followed the same trend as RNA expression. Enzymatic activity in both Nigerian and South African venoms (19.9 ± 1.1 U/I and 18.1 ± 2.8 U/I respectively) was significantly greater than that of the Tanzanian *B. arietans* (5.9 ± 0.1 U/I, $P < 0.0001$ for both).

To determine whether the observed protein and enzymatic variation has a direct impact on phenotypic venom toxicity, MTT assays were performed using human keratinocyte cell lines to assess cytotoxicity (Fig 4a). Significant differences in the potency of the three venoms were observed (Fig 4b), with Tanzanian *B. arietans* venom being the most toxic to HaCaT cells ($IC_{50}$: 17.7 µg/µL ± 3.3), followed by the South *African B. arietans* venom ($IC_{50}$: 27.8 µg/µL ± 2.8), and finally, Nigerian *B. arietans* venom ($IC_{50}$: 37.2 µg/µL ± 2.3). Both Tanzanian

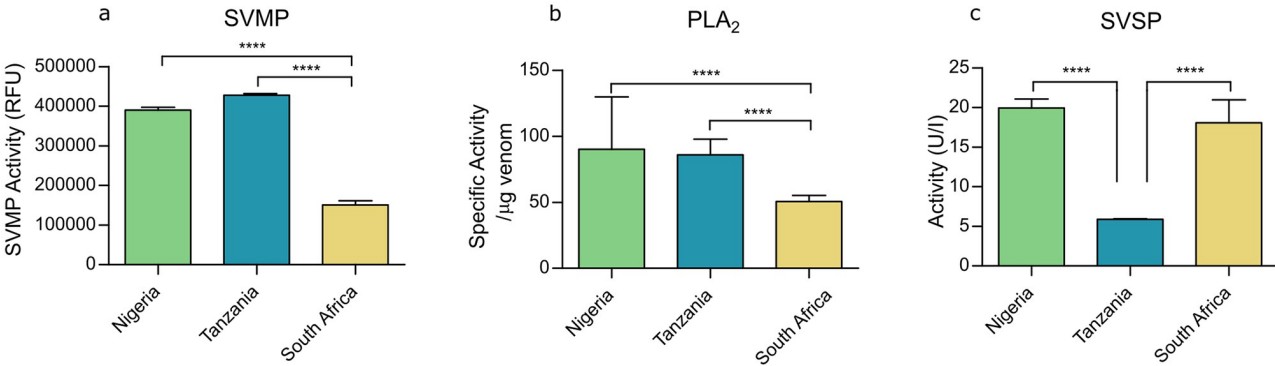

**Fig 3. Enzymatic activity assays show significant intraspecific differences in *Bitis arietans* venom.** Three enzymatic assays were used to quantify snake venom metalloproteinase (SVMP), phospholipase $A_2$ ($PLA_2$), and snake venom serine proteases (SVSP) activity in venoms. a) For SVMP activity, venom activity was assayed with the fluorogenic substrate ES010. Venoms were used at 1 µg/well, in triplicate. Fluorescence was read at an excitation wavelength of 320nm and an emission wavelength of 405 nm for 1 hour. At 1 hour, the fluorescence values were taken as end-read values for statistical comparison, and plotted ± SD. b) For $PLA_2$ activity, venoms were tested in triplicate, at 10 ng/well using a commercially available Abcam secretory $PLA_2$ kit. Absorbance was read at 405 nm and specific $PLA_2$ activity was calculated as per manufacturer's instructions and plotted ± SD c) SVSP activity was measured using the commercially available chromogenic S-2288 substrate. Venoms were used in triplicate at 0.066 mg/mL. Absorbance was read at 405 nm. Data was then used to calculate activity in U/I, plotted ± SD. Quantified activity was then statistically compared between different locales using a Two-way ANOVA with Tukey's multiple comparison tests. Significance is denoted by *, with * P = 0.05, ** P = 0.005, *** P = 0.0005.

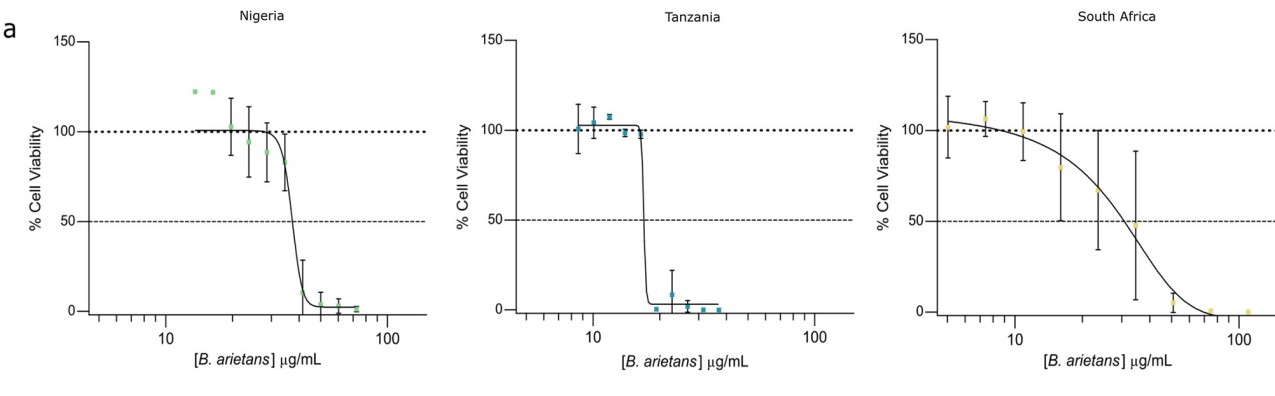

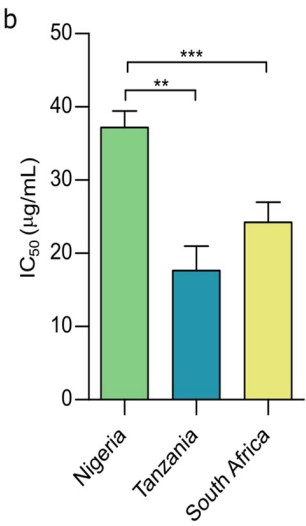

**Fig 4. Venoms from *Bitis arietans* show different cytotoxic potencies to human keratinocytes.** Cell metabolic (MTT) assays were used to measure cell viability 24 hours after exposure to *B. arietans* venom sourced from three different locales. HaCaT cells were seeded at 20,000 cells/well and exposed to serial dilutions of *B. arietans* venom for 24 hours before Thiazole Blue tetrazolium bromide was used to quantify cell viability using a BMG Clariostar plate reader. Data was used to generate dose-response curves for $IC_{50}$ calculations. $IC_{50}$ values were then statistically compared using Two-way ANOVA with Tukey's multiple comparison test. Each experiment was performed on three independent occasions with triplicate readings collected for each condition on the plate. The data shown represent means of triplicate experiments and error bars represent standard deviations. Significance is denoted by *, with * P = 0.05, ** P = 0.005, *** P = 0.0005.

and South African venoms exhibited significantly greater cytotoxic potencies than Nigerian *B. arietans* venom (P = 0.003 and P = 0.031, respectively).

### 3.4 Three clinically relevant antivenoms show differential recognition of variable *B. arietans* venom toxins

End point ELISA and western blotting experiments were used to determine whether geographical variation in *B. arietans* venom composition affected toxin recognition by three polyvalent antivenoms. For ELISA experiments, to compare the extent of venom recognition between the three antivenoms, a discriminatory dilution point of 1 in 62,500 was selected. Highest $OD_{405}$ values were obtained from the incubation of SAIMR Polyvalent against the three *B. arietans* venoms, and there was minimal difference between the $OD_{405}$ values for Nigerian, Tanzanian and South African *B. arietans* venoms ($OD_{405}$ = 1.14 ± 0.09, 1.13 ± 0.12 and 1.20 ± 0.09 respectively) (Fig 5a). Fav-Afrique also showed minimal difference between OD values observed for

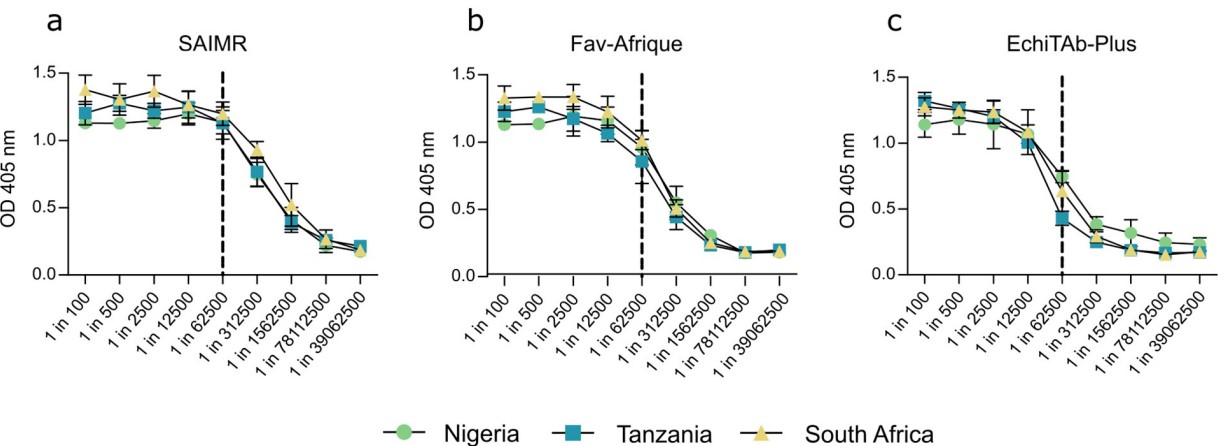

**Fig 5. Intraspecific venom variation has minimal impact on *Bitis arietans* venom recognition by polyvalent antivenoms measured by end-point titration ELISA.** *B. arietans* venom was adsorbed to ELISA plates at 100 ng/well and before being incubated with serial dilutions of three medically relevant antivenoms (SAIMR Polyvalent, Fav-Afrique, and EchiTAb-Plus; dilution range: 1:100–1:39,062,500). Absorbance was measured at 405 nm and plotted ± SD to visualise dilution curves. Dotted line represents comparison point selected, at 1 in 62,500. Graphs prepared using GraphPad Prism 9.

Nigerian, Tanzanian and South African ($OD_{405}$ 0.96 ± 0.13, 0.86 ± 0.16 and 1.01 ± 0.07 respectively) (Fig 5b). Finally, EchiTAb-Plus did reveal differences between $OD_{405}$ values, as recognition of Nigerian venom ($OD_{405}$ = 0.74 ± 0.04) and South African venom ($OD_{405}$ = 0.64 ± 0.15) was markedly greater than recognition of Tanzanian *B. arietans* venom ($OD_{405}$ = 0.43 0.05) (Fig 5c).

Quantitative western blotting was applied to visualise the ability of all three antivenoms to recognise constituent venom toxins (Fig 6). To remove any potential variation in signal due to differences in the IgG concentration of the three antivenoms, all antivenoms were standardised to 10 mg/mL. Venom recognition by the three different serotherapies was quantified in RFUs, with higher values indicating greater toxin recognition. Across all three serotherapies, Fav-Afrique had lower RFU values against all three geographical venom variants. Greater disparity was seen between the RFU values when the three *B. arietans* venoms were probed with SAIMR Polyvalent, with South African *B. arietans* having the highest RFU value (3.17), followed by Tanzania (1.91) and Nigeria (1.96). When venom recognition with EchiTAb-Plus was quantified, South African *B. arietans* produced the highest RFU values (4.63), followed by Tanzanian (1.85) and then Nigerian venom (0.63).

## 3.5 Intraspecific venom variation does not substantially impact antivenom neutralisation of venom activity

To determine potential treatment implications associated with the intraspecific venom variation observed in *B. arietans*, antivenoms were used in *in vitro* enzymatic assays to assess toxin inhibition (Fig 7). All antivenoms resulted in significant reductions in SVMP activities across all three venoms (Fig 7a). EchiTAb-Plus consistently produced the greatest reduction in SVMP activity, resulting in 96.7%, 91.9% and 84.2% reductions against South African, Nigerian, and Tanzanian venoms, respectively. Fav-Afrique inhibited SVMP activities to comparable extents across Nigerian, South African, and Tanzanian venoms (85.0%, 84.6%, and 70.2% reduction, respectively). SAIMR Polyvalent antivenom inhibited Tanzanian and Nigerian venom to a significantly greater extent than the South African samples (82.6%, 78.1% and 64.4% reductions, P = 0.003 and P = 0.0006 respectively).

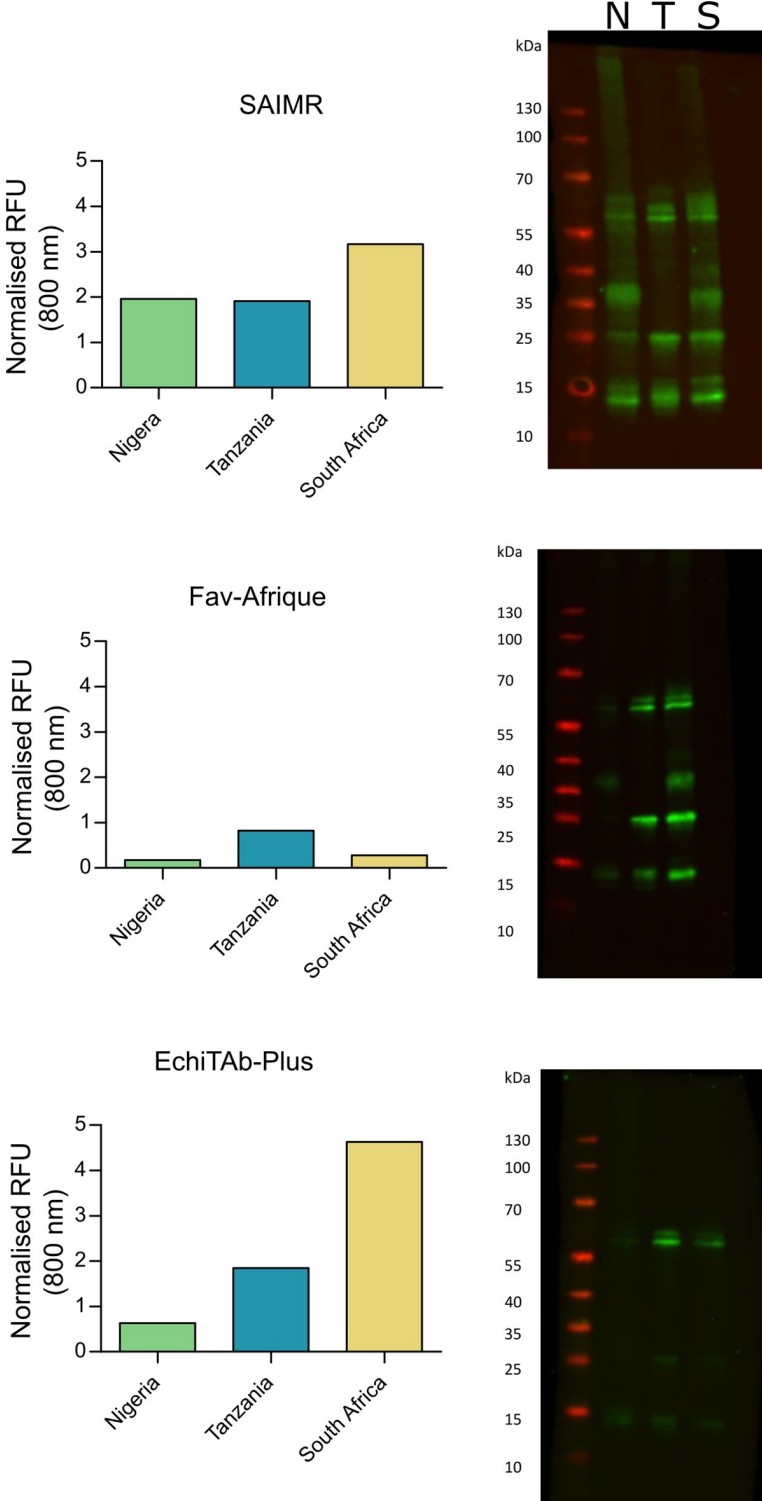

**Fig 6. Immunoblotting of three polyvalent antivenoms against three different geographical variants of *B. arietans* venom.** *B. arietans* venoms were run on pre-cast SDS-PAGE gels under reduced conditions at 1 μg/lane before being transferred to nitrocellulose membranes for immunoblotting. Membranes were incubated overnight with antivenom before being visualised for quantification via a LiCor Odyssey System at 800 nm.

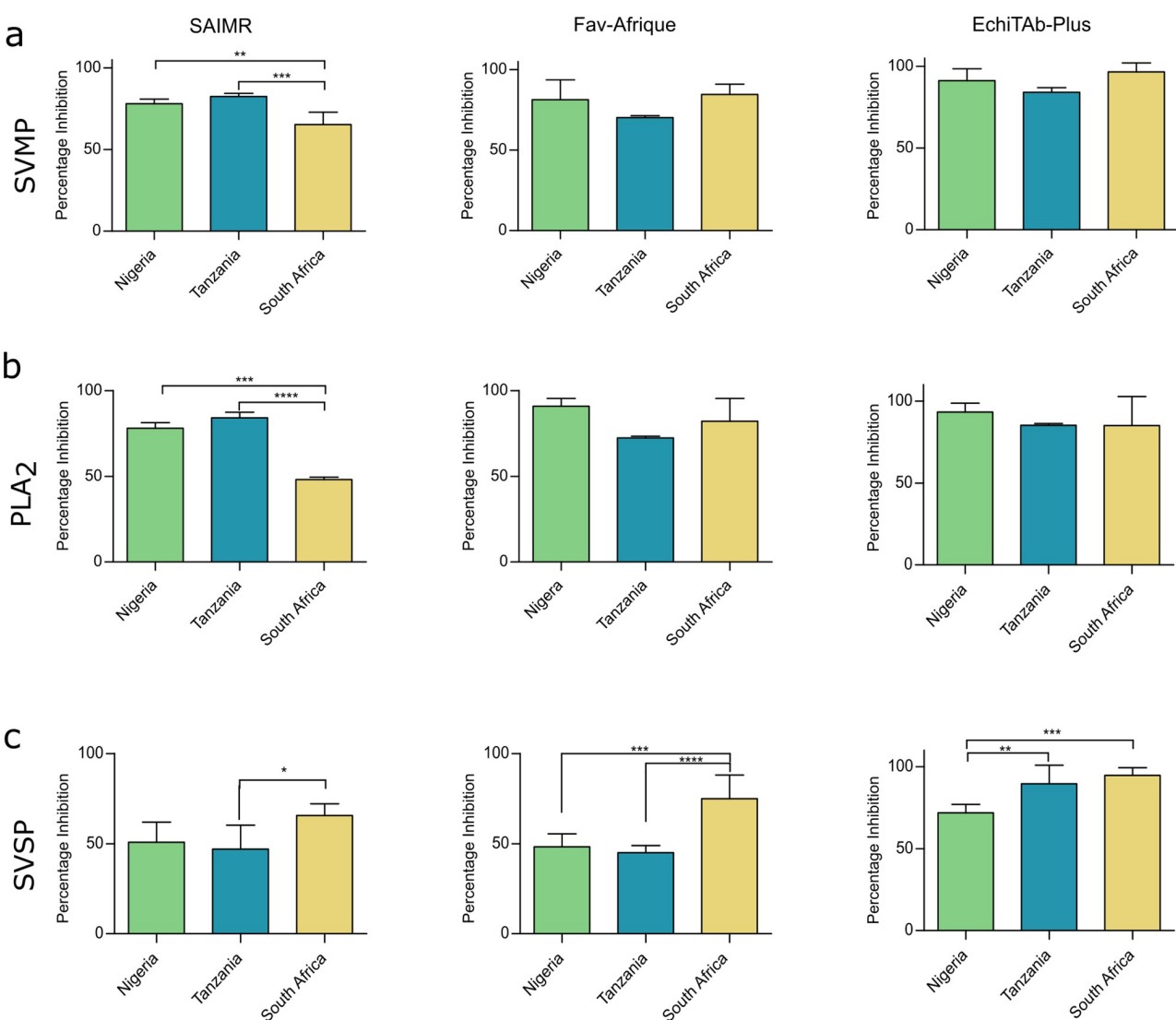

**Fig 7. Antivenoms broadly inhibit *Bitis arietans* venom *in vitro* despite intra-specific venom variation.** The inhibitory capability of three polyvalent antivenoms, SAIMR, Fav-Afrique, and EchiTAb-Plus, against the enzymatic activity of (a) SVMP (b) PLA$_2$ and (c) SVSP was assessed against the three geographically distinct *B. arietans* venoms. The data shown are percentage inhibition calculated relative to a venom only control and represents the mean of triplicate measurements captured over three replicate experiments. Error bars represent ± SD. Inhibition data was statistically compared using a Two-Way ANOVA with Tukey's multiple comparison tests to determine statistical significance in the inhibitory activity of each antivenom between the three locales. Significance is denoted by *, with * P = 0.05, ** P = 0.005, *** P = 0.0005.

Examining the inhibition of PLA$_2$ activity also revealed that the three antivenoms were broadly comparable (Fig 7b). EchiTAb-Plus exhibited highly comparable toxin inhibition, ranging from 85.0% (South Africa) and 85.2% (Tanzania) to 93.4% (Nigeria). Similarly, pre-incubation with Fav-Afrique resulted in percentage inhibition values ranging from 72.4% (Tanzania) to 90.9% (Nigeria). Only SAIMR Polyvalent exhibited significant differences in PLA$_2$ inhibition, with 78.0 and 84.2% reductions in activity observed with Nigerian and Tanzanian venoms, respectively, while the inhibitory potency against the South African venom was significantly reduced compared to the other two locales (Fig 7b, P = 0.0002 and P < 0.0001).

Lastly, the ability of the three serotherapies to inhibit venom SVSP activity was measured. Pre-incubation of the three *B. arietans* venoms with EchiTAb-Plus antivenom had the greatest inhibitory effect against venom from South Africa (94.7%), followed by Tanzania (89.5%) and Nigeria (71.9%) samples (Fig 7c). The inhibition of SVSP activity was significantly greater against both South African and Tanzanian venoms compared to the Nigerian venom (P = 0.0047 and P = 0.0002 respectively). Inhibition by FavAfrique was significantly higher against South African *B. arietans* (79.4%) compared to both Tanzanian (45.0%, P < 0.0001) and Nigerian venoms (48.2%, P = 0.0003). This pattern of inhibition was also observed with SAIMR Polyvalent, which was also more effective against South African *B. arietans* venom (South Africa, 65.6%; Tanzania, 47.0%; Nigeria, 50.8%), though this difference was only statistically significant for the South African and Tanzanian comparison (P = 0.022).

## 4. Discussion

The extensive geographic range of *B. arietans*, coupled with its medical importance in the context of snakebite, makes it an ideal model to study intraspecific venom variation. Previous studies have shown that there is significant variation at both the protein [5] and functional level [22,31] for *B. arietans* venom. Building upon this existing literature, here we applied a combined approach consisting of transcriptomic, chromatographic, enzymatic, immunological, and cellular analyses to samples sourced from three different geographic locales: Nigeria, Tanzania, and South Africa. In addition to these analyses, the impact of the detected venom variation on the *in vitro* performance of three medically relevant antivenoms was also explored.

Quantifying the gene expression of toxin families detected in the venom gland transcriptomes revealed variation between the three locales. Whilst the small number of *B. arietans* specimens used in the study (6 individuals from Nigeria, 3 from Tanzania, and 1 from South Africa) limits the interpretation of data for whole populations, they have revealed extensive variation, nonetheless. The assembled venom gland transcriptomes are in broad agreement with existing 'omics' analyses on *B. arietans*. Previous studies have found that SVMPs, SVSPs and PLA$_2$s constitute the major enzymatically active toxins in this venom [20,30] and the abundance of CLPs detected in Nigerian *B. arietans* has previously been quantified at 44% of all venom toxins [38], supporting the finding here of these being the dominant toxin type expressed at both transcriptomic and proteomic level.

A prior study has shown a large discrepancy between transcriptomic expression and proteomic abundance of toxins detected in *B. arietans* [7]. For the three locales studied here, CLP abundance at both protein and transcript level was around 30%. However, quantification of SVMPs through RP-HPLC showed a greater relative protein abundance compared to that predicted by gene expression. As stated previously, mass spectrometry was not applied for this study, but would be a valuable next step for a more detailed exploration of the protein profiles and isoform identification of the diverse toxins found within these venoms. It has previously been documented that mRNA translation within the venom gland is controlled by pre- and post- translational mechanisms that regulate the final venom protein content [7,54]. In addition to these post-translational mechanisms of gene control, changes in the presence and abundance of toxin genes cannot be ruled out by this study. Within the Nigerian *B. arietans* toxinome, a lack of CRISP mRNA was observed. This lack of expression could indicate gene loss within this individual, however, genome sequencing would be required to test this hypothesis.

Enzymatic variation between different *B. arietans* populations has also been reported in the literature [5]. For example, zymography experiments have previously shown that Nigerian *B.*

*arietans* venom displays greater protease activity than Tanzanian venom, which was hypothesised to be due to differences in SVMP activity [5]. However, in this study we found significantly lower SVMP activity in Nigerian *B. arietans* compared to that observed in Tanzanian specimens. Significantly greater SVSP activity was detected in Nigerian venom compared to the Tanzanian venom though, and the previous zymographic observations may be a result of this, although further characterisation work will be required to confirm or disprove this hypothesis. Enzymatic assays, coupled with the RP-HPLC analyses performed, provided another comparison against the transcriptomic data. For all three assays (SVMP, SVSP and PLA$_2$), the locale with the highest enzymatic activity correlated with the highest toxin transcript expression. Whilst it was only possible to quantify the presence of CLPs and PII-SVMPs through RP-HPLC, the high prevalence of CLPs at the transcriptomic level was reflected in their abundance at the protein level.

Detailed observations of the local clinical manifestations of *B. arietans* envenoming remain underreported, so it is currently unclear whether the venom variation detected here impacts upon the pathology observed in snakebite patients across different parts of Africa. Using a cell death assay with human keratinocytes, we found that the three venoms showed significantly different cytotoxic potencies. Whilst measures of cytotoxicity provided against a monolayer of cells limits the conclusions that can be drawn from this data, our analyses nonetheless suggests that Tanzanian *B. arietans* venom may be more potent compared to venom from either Nigerian or South African *B. arietans*. This possibility requires additional testing, in either a more complex preclinical *in vivo* model or via clinical observations.

Several studies have shown that venom variation can have a considerable impact on the efficacy of antivenom therapy [33,34]. However, previous studies comparing the efficacy of EchiTAb-Plus against *B. arietans* venom sourced from Nigeria and Cameroon showed that there was no impact on antivenom efficacy in a murine model of envenoming [35] In this study, we observed substantial differences in venom recognition by EchiTAb-Plus antivenom in immunoblotting, showing that the venom used as the immunogen (from Nigeria) was recognised to a greater extent than South African and Tanzanian *B. arietans* venom. Nonetheless, the additional immunological recognition of Nigerian *B. arietans* venom did not translate into significant differences between the three locales in the inhibition of enzymatic SVMP and PLA$_2$ activity. Furthermore, when the inhibition of SVSP activity was determined, EchiTAb-Plus had a significantly lower inhibition of Nigerian *B. arietans* compared to both South African and Tanzanian venoms, however it cannot be determined whether this lower inhibition would affect overall antivenom performance *in vivo*.

Despite being a South African product that showed strong recognition of venom from this locale in the immunological assays, inhibition of venom PLA$_2$ and SVMP activities by SAIMR Polyvalent antivenom was significantly lower against South African venom compared with the venoms sourced from Nigeria and Tanzania. Only SVSP inhibition was significantly greater in the South African venom compared to the other venom samples. Re-examining the functional activity of South African *B. arietans* venom reveals low PLA$_2$ and SVMP activity compared to the other geographical locales, but significantly greater SVSP activity. It could therefore be hypothesised that inhibition of this toxin class could be the most important when considering how these antivenoms may perform *in vivo*, particularly when considering the reported high potency of this antivenom against *B. arietans* venom in preclinical studies [55,56].

Across all three serotherapies, Fav-Afrique had the lowest recognition of the three geographical venom variants in the immunological assays, which may be related to the use of historical stocks of this antivenom which expired in 2016, given the hiatus in manufacturing of this product since 2010. Despite this, when the ability of Fav-Afrique to inhibit the enzymatic activity of pathologically relevant toxins was explored, it showed strong inhibition of SVMP

and PLA$_2$ activity against venom from all three *B. arietans* locales. Antivenom efficacy extending beyond the manufacturer's expiry data has been documented for other products, and it is promising to observe the same regarding Fav Afrique in an *in vitro* context [57]. Only inhibition of SVSP activity varied significantly, with Fav-Afrique causing a significantly lower inhibition of SVSP activity caused by Nigerian and Tanzanian *B. arietans* venom compared to South African venom.

Taking the results presented here and the existing literature into account, the medically important snake species *B. arietans* exhibits clear and measurable intraspecific venom variation; however, the pathophysiological consequences of this variation remain unclear. To further evaluate how compositional variation might affect venom potency, more complex experimental models than enzymatic assays are required, with both hen's eggs and murine *in vivo* assays offering a robust vascularised environment for assessing the haemotoxic and cytotoxic activities of snake venoms, and for better evaluating the neutralising efficacy of antivenoms against these pathologies.

Finally, despite the medical importance and broad geographical range occupied by *B. arietans*, there remains a lack of comprehensive clinical case series from different African regions relating to envenoming caused by this species. Such research is fundamental to understanding the real-world consequences of *B. arietans* venom variation in human snakebite patients. Despite these knowledge gap-associated limitations, our study highlights that different geographical populations of *B. arietans* exhibit variation in venom toxin composition, but that this variation may only have modest consequences to *in vitro* venom function and antivenom neutralisation. This fundamental research strongly advocates for additional research in this space to better understand the causes and consequences of intraspecific venom variation across the geographical ranges of medically important snake species.

## Supporting information

**S1 Fig. SDS-PAGE analysis of the CLP-containing peaks in the RP-HPLC profiles of B. arietans venoms.** Fractions from the HPLC traces shown in Fig 2 in the main text were subject to SDS-PAGE under reducing conditions and the those containing a pair of bands in the 13–18 kDa region typical of alpha and beta CLP subunits are shown [left hand gels]. These same fractions were also subjected to SDS-PAGE under non-reducing conditions [right hand gels] and most contained a strong band at 28–29 kDa corresponding to the intact heterodimeric CLP. In some minor forms [TZA 3, NGA 2 and SA 4] a 70–75 kDa protein was observed under non-reducing conditions and these are likely to be high molecular weight forms of CLPs. The SVMP peaks were identified by co-elution of the respective SVMPIIs purified in a separate study [52]. The gels used were BioRad 4–20% acrylamide, stained with Coomassie Blue R250.
(DOCX)

## Acknowledgments

We thank Paul Rowley and Edouard Crittenden for the maintenance and husbandry of the snake collection and the provision of venom samples at LSTM.

## Author Contributions

**Conceptualization:** Charlotte A. Dawson, Stuart Ainsworth, Robert A. Harrison, Nicholas R. Casewell.

**Data curation:** Charlotte A. Dawson, Keirah E. Bartlett, Mark C. Wilkinson, Cassandra M. Modahl.

**Formal analysis:** Charlotte A. Dawson, Keirah E. Bartlett, Cassandra M. Modahl.

**Investigation:** Charlotte A. Dawson, Keirah E. Bartlett, Mark C. Wilkinson, Laura-Oana Albulescu, Taline Kazandijan, Adam Westhorpe, Rachel Clare, Cassandra M. Modahl.

**Methodology:** Charlotte A. Dawson, Keirah E. Bartlett, Mark C. Wilkinson, Stuart Ainsworth, Laura-Oana Albulescu, Taline Kazandijan, Steven R. Hall, Adam Westhorpe, Rachel Clare, Cassandra M. Modahl.

**Resources:** Simon Wagstaff.

**Software:** Simon Wagstaff, Cassandra M. Modahl, Robert A. Harrison.

**Supervision:** Stuart Ainsworth, Laura-Oana Albulescu, Cassandra M. Modahl, Nicholas R. Casewell.

**Validation:** Charlotte A. Dawson.

**Visualization:** Charlotte A. Dawson, Keirah E. Bartlett, Steven R. Hall.

**Writing – original draft:** Charlotte A. Dawson, Nicholas R. Casewell.

**Writing – review & editing:** Charlotte A. Dawson, Keirah E. Bartlett, Mark C. Wilkinson, Stuart Ainsworth, Laura-Oana Albulescu, Taline Kazandijan, Steven R. Hall, Adam Westhorpe, Rachel Clare, Cassandra M. Modahl, Robert A. Harrison, Nicholas R. Casewell.

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
