## [Decision Letter · Decision Letter 0]

24 Jul 2024

Dear Dr Dawson,

Thank you very much for submitting your manuscript "Intraspecific venom variation in the medically important puff adder (Bitis arietans): comparative venom gland transcriptomics, in vitro venom activity and immunological recognition by antivenom" for consideration at PLOS Neglected Tropical Diseases. As with all papers reviewed by the journal, your manuscript was reviewed by members of the editorial board and by several independent reviewers. The reviewers appreciated the attention to an important topic. Based on the reviews, we are likely to accept this manuscript for publication, providing that you modify the manuscript according to the review recommendations. 

The reviewers have now finished evaluating the manuscript. They addressed some important suggestions, mainly regarding the transcriptome assembly and analysis. We suggest the authors consider these comments and prepare a revised manuscript version.

Sincerely,

Ana Moura-da-Silva

Academic Editor

José María Gutiérrez

Section Editor

The reviewers have now finished evaluating the manuscript. They addressed some important suggestions, mainly regarding the transcriptome assembly and analysis. I suggest the authors consider these comments and prepare a revised manuscript version.

Reviewer's Responses to Questions

**Key Review Criteria Required for Acceptance?**

**Methods**

-Are the objectives of the study clearly articulated with a clear testable hypothesis stated?

-Is the study design appropriate to address the stated objectives?

-Is the population clearly described and appropriate for the hypothesis being tested?

-Is the sample size sufficient to ensure adequate power to address the hypothesis being tested?

-Were correct statistical analysis used to support conclusions?

-Are there concerns about ethical or regulatory requirements being met?

Reviewer #1: Methods are appropriate to address the objectives.

Reviewer #2: Overall I have no major concerns with the methodology. A minor concern is the choice of assembly methods which may impact results (see below).

Reviewer #3: Overall the authors performed a robust analysis given the available data. My main focus is going to be on the transcriptomic analyses. With that said, my main concern is that Authors do not specify if they checked for chimeric contigs using any kind of coverage approach. This step is crucial as it removes artefactual contigs that appear to be real toxins. If they did check for chimeric transcripts it is relevant to mention it. If they did not, they should consider doing it and reanalyzing the data if the analysis removes a lot of the previously identified contigs. This is relevant as on the results the authors compare the number of retained contigs for several toxin families. These numbers could be impacted by doing a chimeric contig analysis. All other methods are clearly stated and are robust in my opinion, although I am more knowledgeable in transcriptomics and will focus my reviews in this aspect of the paper.

**Results**

-Does the analysis presented match the analysis plan?

-Are the results clearly and completely presented?

-Are the figures (Tables, Images) of sufficient quality for clarity?

Reviewer #1: The results are clearly presented and the figures are of sufficient quality.

Reviewer #2: The results are well presented. I wonder if the low recovery of CRISPs could be due to some failure in assembly. These can be difficult and often why some authors choose to use multiple assembly methods (see Holding et al 2018 Toxins).

Reviewer #3: Results are clear and concise. The authors structured the paper in a magnificent manner and it is super easy to read and comprehend. However, as mentioned earlier they brought a lot of focus to the number of retained contigs for the predominant toxin families in each individual. Doing this without a chimeric contig analysis is, in my opinion, not recommended as those numbers can be heavily misleading after accounting for chimeras. Authors should state whether or not they performed that analysis, and in case they did not perform it, they should reanalyze the retained transcripts to search for chimeras and remove them for subsequent analyses. Also, I think the authors could deepen a bit into those differences in the number of contigs. For example: How many of the retained CLPs are unique to each population? It might be interesting to see if each population is using the same genes or if they are expressing different alleles. Maybe provide the CDSs of the annotated CLPs for each individual? These applies to all relevant toxin families, and could bring an evolutionary context for the three populations, in which we would have at least an indication of which and how many genes are being selected in each population and if there are noticeable differences in the core number of expressed paralogs among them. Also, a general comment is that authors are claiming that some toxins have substantially lower or higher expression levels using only the Toxinome. This can be extremely misleading as they do not account for significant (and rather common) differences in toxin proportions when the total transcriptome (including non-toxins) is considered. I would not make such claims unless I analyze the total transcriptomic context of the venom gland. Considering the example provided at the end of the paragraph between lines 324-332: That 1.5% difference in SVMP PIII expression between Nigerian and Tanzanian individuals could be completely non-existent or even reversed when analyzing the whole transcriptome. PIII could be harboring a higher proportion of reads in the Nigerian individual. That is why these comparisons should be made using the non-toxin context of the whole transcriptome, which represents more clearly the actual phenotype of the venom gland. In my opinion, authors should either refrain from stating those differences in expression or perform the comparison using the whole transcriptome for a more confident analysis. Other results are well organized and clearly stated with good figures. Again, I would like to congratulate the authors in this regard.

**Conclusions**

-Are the conclusions supported by the data presented?

-Are the limitations of analysis clearly described?

-Do the authors discuss how these data can be helpful to advance our understanding of the topic under study?

-Is public health relevance addressed?

Reviewer #1: The conclusions are supported by the data. The authors emphasize the manuscript's contribution to the topic, addressing its public health relevance.

Reviewer #2: The conclusions are well justified as is the public health relevance.

Reviewer #3: Conclusions are generally well supported by the data although Authors should consider my comments on their transcriptomic analyses to have a more robust claim when stating significant differences in toxin expression among individuals. Also, they could include more evolutionary details comparing the retained contigs for relevant toxins and checking if they represent different or conserved genes. I only mention this because they do make an interesting insight regarding the absence of CRiSPs in the Nigerian individual, which might indicate a gene-loss event in that population. Their dataset allows for super interesting evolutionary hypotheses of gene selection among the three populations which might also have relevant medical implications if significant differences are found. I would find that discussion super interesting. I would like to confirm if the authors checked for chimeras on each individual and if they did not I recommended them doing so to avoid any misleading sequences being retained in the final datasets. Finally, they should reanalyze their expression data accounting for non-toxin transcripts to have a more robust claim on which toxins are highly or lowly expressed in the venom gland of each individual. Making these comparisons using only the universe of toxins can be misleading and lead to incorrect conclusions. Other than that, authors provided a well-structured manuscript with relevant results. Congratulations to all involved for the hard work. It was a pleasure reading your work!

**Editorial and Data Presentation Modifications?**

Reviewer #1: (No Response)

Reviewer #2: Accept

Reviewer #3: I am marking this paper as needing only minor revisions because all the analyses I recommended are easy and simple to perform as authors only have three samples to analyze. However, I do believe that, without a strong justification for not following my recommendations, authors need to perform them in order to publish a more robust and high-quality work. This goes only for the transcriptomic part of the paper. All other analyses are robust and clearly explained in the paper, and the conclusions are fitting as well. Overall, it is a marvelous work!

**Summary and General Comments**

Reviewer #1: The manuscript describes the intraspecific variability of Bitis arietans venom from three different locations (Nigeria, Tanzania, and South Africa), as well as the immune recognition and neutralisation of its three main enzymatic activities (SVMP, SVSP, and PLA2). The manuscript is well-grounded, justifying the relevance of studying the compositional and functional variability of venom from species with broad geographic distribution and significant medical importance. The high-quality results are well presented and discussed. I recommend publishing this manuscript in PLOS Neglected Tropical Diseases after revision, as outlined in the attached file.

Reviewer #2: This is a relatively straightforward but thorough study of the geographical variation in venom profile and antivenom efficacy in B. arietans populations from multiple localities (though certainly not range-wide).

Reviewer #3: (No Response)

PLOS authors have the option to publish the peer review history of their article (what does this mean?). If published, this will include your full peer review and any attached files.

Reviewer #1: No

Reviewer #2: No

Reviewer #3: No

Figure Files:

Data Requirements:

Reproducibility:

References

---

## [Editor Report · Decision Letter 1]

24 Sep 2024

Dear Dr Dawson,

We are pleased to inform you that your manuscript 'Intraspecific venom variation in the medically important puff adder (Bitis arietans): comparative venom gland transcriptomics, in vitro venom activity and immunological recognition by antivenom' has been provisionally accepted for publication in PLOS Neglected Tropical Diseases.

Best regards,

Ana Moura-da-Silva

Academic Editor

José María Gutiérrez

Section Editor

The revised version of the manuscript has taken into account the comments made by the reviewers and has included the suggested corrections to our satisfaction. Therefore, I recommend publishing the manuscript in its current form.

---

## [Editor Report · Acceptance letter]

15 Oct 2024

Dear Dr Dawson,

We are delighted to inform you that your manuscript, "Intraspecific venom variation in the medically important puff adder (Bitis arietans): comparative venom gland transcriptomics, in vitro venom activity and immunological recognition by antivenom," has been formally accepted for publication in PLOS Neglected Tropical Diseases.

Best regards,

Shaden Kamhawi

co-Editor-in-Chief

Paul Brindley

co-Editor-in-Chief
